# Influence of the Profession and Industry of Work on the Labor Mobility of the Applicant

**Alexey Tikhonov [1],\*, Sergey Novikov [1], Vyacheslav Kalachanov [1] and Umberto Solimene [2]**

[1]  Department of Human Resource Management, Moscow Aviation Institute (National Research University),
    Volokolamskoe Shosse 4, 125993 Moscow, Russia; svnovikovmai@mail.ru (S.N.); k506@mai.ru (V.K.)

[2]  Department of Biomedical Sciences for Health, University of Milan, Via L. Mangiagalli 31, 20133 Milan, Italy;
    umberto.solimene@unimi.it

\*  Correspondence: tikhonovmai@mail.ru

**Abstract:** The article examines the problem of the influence of the profession and industry of work of Russian applicants on their labor mobility. The general growth of labor mobility of the population is currently caused by several factors: change in the labor values of applicants, technological progress, desynchronization of the education sector and the labor market, growth of the economic crisis, etc. The main reasons prompting applicants to think about changing their current job in the article are the aspects of their relation to those professional areas and industries in which they are currently working or would like to work in the future. The authors analyzed the results of surveys of applicants of various ages and from various professional fields regarding their desire to change their profession (without taking into account the influence of the material factor), as well as their opinions regarding the most attractive professional fields for them. In addition, there are the opinions of applicants regarding the reasons prompting them to think about changing their profession. The article also examines data from interviews with applicants regarding their desire to move to work in a company from another industry.

**Keywords:** labor mobility; profession; industry

## 1. Introduction

One of the important trends in the development of the modern labor market in Russia and around the world is to increase the labor mobility of the population. Several major factors, both global and specific to Russia, contribute to this mass phenomenon. Among them there is a change in the labor values of applicants, accelerated technological progress, disrupted interaction between education and labor market and increase in job dissatisfaction among young professionals (Tikhonov and Novikov 2020).

Sociological studies show that new generations of applicants are more inclined to change jobs than older specialists. At the moment, the average young applicant in the United States is expected to change 11 jobs during his life (Chernikov 2014). Changes in the labor values of new generations of workers are also observed in Russia. In particular, among young applicants in the Russian labor market, there is an importance of such values as professional development in the company, interesting job responsibilities, recognition and sense of significance at work, comfortable atmosphere in the work team and flexible work. Thus, young applicants, in general, turn out to be more demanding of their work and less willing to adapt to their current employer for the sake of stability or high income, preferring the strategy of finding a suitable job for them, which contributes to their horizontal mobility.

The situation of changing labor values is especially noticeable in high-tech spheres, such as aerospace industry, as well as those related to creative work. The values of qualified information technology (IT) workers are most different from the average for the Russian labor market and are more

similar to the values of young applicants in developed countries. One of the reasons may be the influx of young and flexible employees into companies from the information sector. For example, in the field of software development, there is a paradoxical situation of a low entry threshold: for employment in a junior developer position, a specialized education is often not required, and the presence of specific skills is checked by the employer using a portfolio of projects or a test task. Together with the high salaries of developers (which, however, usually concern middle and high-level specialists, but not beginners), this attracts specialists from other fields and the trajectory of the transition to work in the field of IT has become the typical one on the Russian labor market.

The situation of lack of demand for higher education as a whole turns out to be typical for many industries: according to statistics, in some regions of Russia more than a half of university graduates do not consider it important for themselves to work in their specialty (Pakina 2014). Although the very existence of a higher education diploma continues to be important for recruiters, the qualitative gap between the demand of the labor market and the supply of universities leads to the fact that the very content of education in some professions is partially devalued, and technological progress only accelerates it (Popova 2018).

The modern pace of industrial development has changed the very approach to education in some areas of work. If previously the typical career path of an employee involved a one-time education and subsequent work in this area with possible stages of advanced training, today in high-tech companies they expect constant and continuous training from employees throughout the entire period of work: this situation is observed in the already mentioned area of software development, where technology obsolescence occurs extremely quickly and some specialists have to change the pool of skills on average every 18 months (Pakina 2014). All this also leads to an increase in the flexibility of employees and their willingness to move to work in a company that is unlike the current position, even while maintaining the current profession.

Finally, another factor affecting the horizontal mobility of applicants is employee dissatisfaction with the nature of work. Despite the lack of sociological research in this area, there is evidence of growing dissatisfaction with their work among young professionals in developed countries, who more often begin to consider their work meaningless and not bringing real benefit to society, even if it is well paid (Antel 2005). This encourages workers in such spheres (for example, financial sphere is mentioned in this regard) to look for the best use of their skills and abilities and change jobs, guided by the values of personal happiness, while losing part of their wealth. As it will be shown below, in Russia there are also certain professions and industries whose employees feel less happy in the workplace and would like to change jobs more often, and the financial sector is one of them (Walsh and Volini 2017).

The aim of the research is to study such a phenomenon, which is quite new for Russia, as labor mobility of the population. The authors conducted a literary review on the topic under study, ranging from the works of the founders of economic theory to modern scientific research by Russian and foreign scientists. The theoretical and methodological method of the research is substantiated. On the basis of sociological and economic research, a scientific and practical analysis of the choice of certain professions by the population as a factor of labor mobility has been carried out. The share of specialists in the Russian labor market who are dissatisfied with the existing profession and are ready to change it to a new job is estimated. A rating analysis of various industries was carried out: from priority to undesirable. The reasons for the attractiveness of industries for job seekers are considered. The influence of such factors as education and age of the employee on labor mobility has been studied. The most favorable industries for work are indicated as conclusions. This data can be used in practical work by the heads of organizations: employers and representatives of recruitment agencies.

## 2. Literature Review

Research issues of labor mobility of the population are widely presented in research and analytical works of various authors, as well as in the reports of consulting companies and HR organizations. In (Chernikov 2014), a definition of the concept of labor values is given, and a retrospective analysis

of the transformation of labor has revealed the dependence of labor values on the historical era, culture, national identity, and employment structure. The author of the study (Popova 2018) classified various factors in the transformation of labor values and showed that changes at the institutional level of management can become the basis for the formation of a new system of values. In work (Gorbunova 2009), the problem of changing professional activity by people of mature age is considered, and on the basis of the analysis of the experimental results the motives of the professional retraining of the individual, mechanisms of the process of transition of workers to new activities are revealed. Greber (2018) wrote a book, in which he shows how our attitude to work came about and how we can improve it. Exploratory research (Kotomina and Shirokshina 2019) aims to examine important aspects of creative retention in the context of generational theory. Managers' understanding of the factors of retention of the most capable employees of Generation Y can form the basis of effective staff policy and minimize the risk of their leaving the organization. The article (Pakina 2014) analyzes the labor behavior and values of modern students, position of labor values in the structure of basic values, attitudes, knowledge, and wishes for a future profession based on the material of a sociological study. In a scientific study (Bing 2011), an empirical analysis of the state of the cross-border flow of labor and its impact on regional economic growth was carried out, the following aspects were considered in detail: choice of the location of the enterprise, regional growth, impact of labor mobility, impact on regional differences in labor mobility and effect of labor mobility of employment and recycling income, and prices. Article of Power and Lundmark (2004) is devoted to the study of labor market dynamics in various industry clusters. British scientists (Green et al. 2000) investigated the influence of various types of training on workers' expectations regarding labor mobility. Unlike most previous research on job change workers, the paper of Ruhm (2007) examines differences in earnings change by gender. Unfortunately, until now, there has been no systematic distribution of industries according to their attractiveness among applicants, which was done by the authors in this work (Akhmetshin et al. 2019).

There are many authors in the world who study the mechanisms of the influence of the profession on the labor mobility of the population. The most notable contribution to the theoretical substantiation of the characteristics of the labor market was made by the famous Scottish economist and philosopher Adam Smith, who is one of the founders of economic theory as a science (Sharma 2020). His follower Ricardo (2012), a classic of political economy, developed a distribution theory that explains how this value of goods is distributed among various professional communities. We used these theoretical conclusions of the past centuries to analyze the current state of income distribution of Russian workers depending on the direction of their labor activity. Keynes (1936) predicted that as productivity rises, working hours will progressively decrease, and creating an enabling environment in which people's lives will become "reasonable, enjoyable and dignified". This postulate is being realized when considering the possibility of switching to four-day work in Russia, and its partially remote nature to improve the complex epidemiological situation caused by the global COVID-19 pandemic. The sociologist Auguste Comte was actively involved in the development of the theoretical and methodological foundations of the study of substantiation of the differentiation of labor activity. In our study, it is the differentiation in terms of wages and working conditions that is the fundamental topic for choosing an actual profession.

The founder of the organic school in sociology, Herbert Spencer, developed the theory of positivism and the general laws of evolution in labor economics. German scientist Max Weber substantiated the theory of rationalization, explaining the general interaction of industry, sociology and culture. American sociologist Talcott Parsons preached the ideas of structural functionalism and founded the theory of human action. All of them investigated the phenomena and patterns taking place in the social and labor sphere that formed the methodological basis for subsequent paradigms. American economist Campbell Macconnell comprehensively analyzed the patterns of development of the labor market and ways of its optimal regulation from the standpoint of various schools and conceptual approaches. The theoretical development of the processes of social mobility of workers and its types was studied by Park (1928) and Stonequist (1961).

A number of authors (Clark 1997; Easterlin 1995; Oswald 1997) have worked to identify differences in satisfaction with the standard of living among workers with different employment status. Richard Freeman viewed job satisfaction as an economic category that can be not only a cause, but also a consequence of employee layoffs. A general idea of the level of satisfaction with the standard of living and work in Russia can be drawn from the results of a study within the framework of the International Social Survey Program, which was carried out on a database for 21 countries (Sousa-Poza and Sousa-Poza 2000). The methodology of the "Bottom-up" theory was used, on the basis of which job satisfaction was assessed taking into account the balance of efforts ("work-role inputs") and results ("work-role outputs") of workers. Russia was the twentieth in terms of the average level of job satisfaction among the 21 analyzed countries. Our research task is to show Russian managers how to improve the situation and raise the Russian economy from the penultimate, not prestigious, place to higher positions.

## 3. Research Method

The basis of the research work carried out is various methods of theoretical and practical research: analysis and generalization, synthesis and abstraction. The authors conducted a comprehensive organizational and economic analysis of job satisfaction in various professional areas of Russia. The research methodology provided for the economic analysis of data from opinion polls of applicants on the Russian labor market conducted by HeadHunter and Deloitte, as well as data from other researchers. At the qualitative level, a classification of the main factors influencing the desire of Russian applicants to change their profession or place of work, depending on their current profession or place of work, has been carried out.

The theoretical and methodological basis of the study was the complex use of a system of methods used by Russian and foreign experts in the field of labor economics: methodology of structural functionalism, systems approach, structuralism, activity approach and interactionism. This required an appeal to a wide range of foreign and Russian concepts that served as methodological guidelines for the authors: theoretical and methodological approaches of classical economics to the system analysis of society and socio-economic systems; conceptual provisions of the stratification theory; theoretical foundations of staff motivation; methodological developments of representatives of psychological and pedagogical science in the field of vocational guidance of workers; research approaches in the field of profession and education; conceptual framework for studying the labor market by economists and researchers in the field of economic sociology and sociology of labor.

In addition to theoretical ones, specific scientific and practical methods of socio-economic research are actively used in research work, first of all, a questionnaire survey, analysis of documentary sources and a method of statistical analysis of statistical information. Information sources used in scientific work include documentary and statistical materials; socio-demographic data on the composition of the population, professional and qualification structure of workers and specialists of enterprises and organizations; materials of sociological surveys of the main categories of the employed and unemployed population of working age; special literature on the topic of research and media materials.

The method of recruiting participants in the study includes a range from 2017 to 2019; the criterion for including recruiting participants was the working age, which in Russia is accepted from 18 to 65 years. The sample under study can be considered representative given the large size of thousands of collectives of workers in various sectors of employment and the long study period.

## 4. Finding and Discussion

### 4.1. Choice of Profession as a Factor of Labor Mobility

The main reason for workers' dissatisfaction with their current profession (as well as the main labor value for Russians in general (Magun 2009)) is the size of their current income (Popova 2018). However, the financial situation is not the only reason for the desire to change the profession: the data

of the interviews of applicants, carried out by HeadHunter, discussed below, are also influenced by such factors as dissatisfaction with work activities and lack of demand in the labor market or poor career prospects (Almanac HeadHunter 2017). Moreover, if the amount of earnings or the demand for specialists in a particular professional field can be called a clear factor of influence (as a rule, there is reliable data on the size of salaries in a particular professional area and on the balance of supply and demand in a certain segment of the labor market), the mood applicants regarding job satisfaction or their prospects can be more hidden for the employer.

The authors put forward the main hypothesis: the majority of employees consciously choose their profession, are able to clearly define their inclinations and preferences, match their interests with their abilities, and have information in the world of professions. Additional hypotheses are the following:

1.  Some employees are not satisfied with their choice of profession;
2.  Not all workers understand whether their specialty is in demand on the labor market;
3.  Most of the young specialists do not know whether they will work in their profession.

According to research data, in general, on the Russian labor market, 10% of applicant changed his profession (Popova 2018). However, there are more people who want to change it for some reason. According to the research (Almanac HeadHunter 2017), 28% of applicants would change jobs if all specialists in the market received the same salaries. According to other studies, more than half of those wishing to change their profession want to do it because of an unsatisfactory financial situation (Gorbunova 2009; Popova 2018). Thus, in the labor market there is not only a large share of specialists who want to change their profession, but also a significant part of applicants who are dissatisfied with the current profession, but continue to work in it in order to save income. Moreover, the share of those wishing to change jobs not because of low income is increasing among younger specialists: for example, according to Almanac HeadHunter (2017), among specialists over 45 years old, only 19% would change jobs if everyone was paid the same, and among specialists under 25 years old there are already 32% (Figure 1).

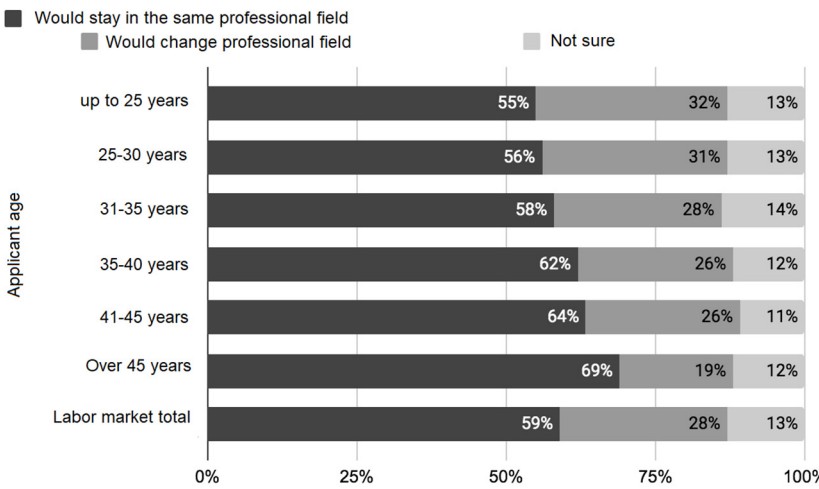

**Figure 1.** Answer of applicants to the question "If everyone was paid the same high salary, you would … " (Compiled by the authors based on materials of Almanac HeadHunter (2017)).

The leader in terms of the share of dissatisfied employees among professional areas in Russia by a large margin is the sales sector: 19% would like to change their profession (Figure 2). The following professional areas in this rating are: finance and economics, accounting and tax accounting (they have at least 7% of employees wishing to change their profession) (Almanac HeadHunter 2017).

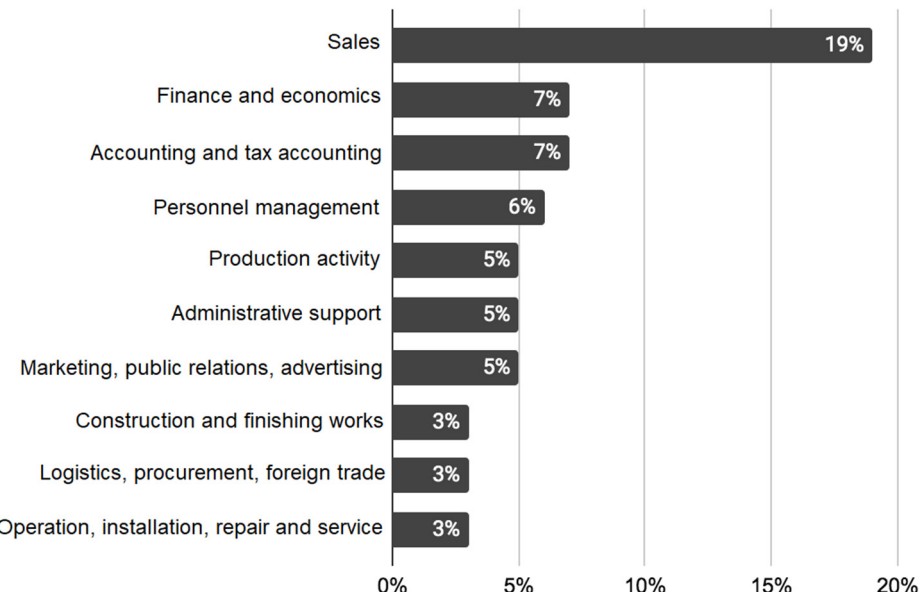

**Figure 2.** Share of applicants who would like to change jobs, subject to the same high salary, in various professional fields (Compiled by the authors based on materials of Almanac HeadHunter (2017)).

The most common reason for wanting to change the professional field (without taking into account the material factor) is the desire to try something new. This was stated by 47% of HeadHunter respondents who, in principle, would like to change their field of activity. Another 37% chose the answer "I have several hobbies, why not", 26% of respondents reported that they were emotionally burned out or exhausted at their current job and that they were bored at their current job. 20% reported that, in his opinion, "having two professions is prestigious". A smaller proportion of respondents stated that they "do not like their current job" (17%). The answer option "only after a while I realized that it was not for me" was chosen by 14%. Only 6% would like to change jobs, as they expect the imminent disappearance of their professional field from the labor market (Almanac HeadHunter 2017).

The reluctance to change the current profession is most often (58%) due to the fact that the employee is satisfied with everything in his professional field. For 30%, the current job is a "dream job". In other cases, reluctance to change jobs is associated with less positive factors. Thus, 14% of respondents are simply too used to their current job, 10% do not have the necessary experience to change their profession and do not know how to acquire it. 5% do not have the necessary education and the desire or willingness to receive it. Only 3% admitted that they lack determination to change their profession and they are too afraid of changes (Almanac HeadHunter 2017).

The most attractive professional area for applicants (still without taking into account the material income factor) is the sphere of art and culture: 12% of those wishing to change their profession would like to work in it. Considering that the main factors in changing it for them is the desire to try themselves in a new field or to realize their hobbies, the choice of this particular professional field looks logical. In second place in the ranking of attractive professional areas was "Human Resources Management": 8% of those wishing to change the type of work would like to move to it (Almanac HeadHunter 2017). A detailed ranking of attractive professional areas is presented below (Figure 3).

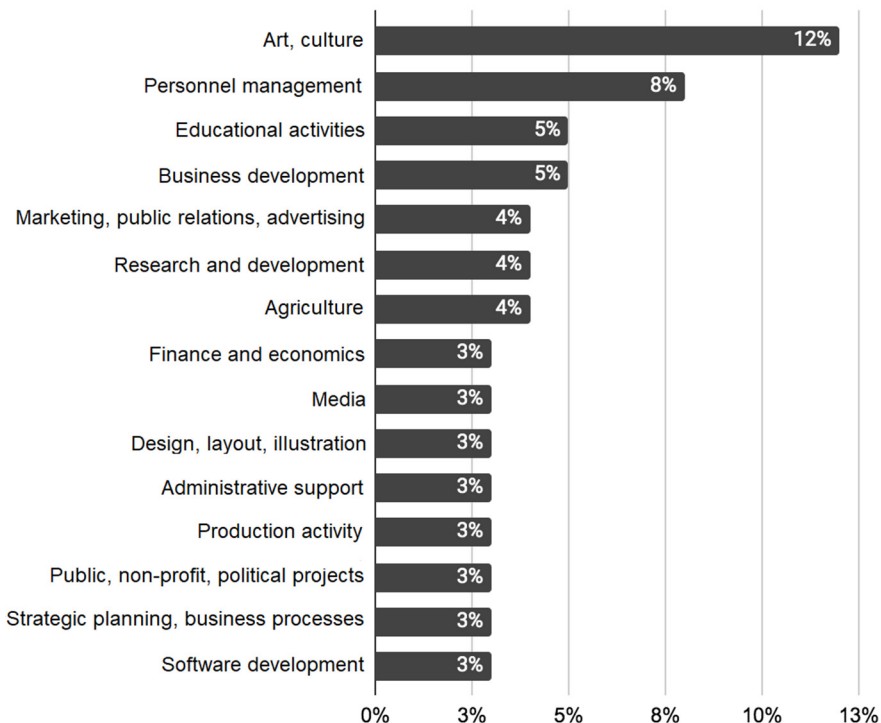

**Figure 3.** Answer of applicants to the question of which profession from which professional field they would change their current job (Compiled by the authors based on materials of Almanac HeadHunter (2017)).

### 4.2. Choice of the Industry of Work as a Factor of Labor Mobility

Although the reasons prompting people to change their profession are varied, it was noted above that the main of them remains money. According to HeadHunter, the two main reasons for voluntary layoffs in companies in the Russian market are the lack of career opportunities and the refusal to raise wages (Almanac HeadHunter 2017). Good salaries are the most frequent criterion by which applicants evaluate the industry as attractive for them (Magun 2009). Thus, the transfer from one industry to another, like the change of profession, is often driven by money. Nevertheless, as the data of surveys of Russian applicants show, this criterion is not the only one among, the work in some industries turns out to be attractive for a significant number of applicants not because of the amount of income, but for other reasons (Kristensen and Westergard-Nielsen 2004).

According to the HeadHunter survey, the most attractive industry for Russian applicants is the IT industry: 20% of applicants would like to work there. Almost the same number of applicants rate the public sector as attractive for them. The third and fourth places in the ranking are shared by the arts and culture, as well as oil and gas (Almanac HeadHunter 2017). As it will be shown below, all of these industries turned out to be the most attractive for applicants for various reasons.

Such prestigious industries as "Hotels, Restaurants, Public Catering, Catering", "Art, Culture" and "Government Organizations" also are in the list of the least attractive industries for applicants: there are many people on the Russian labor market who work in these industries and they are not desirable. However, the undisputed leader of this anti-rating was municipal or urban engineering industry: 33% of interviewed applicant would not want to work there. The main reasons for its unattractiveness are low salaries, conflicts and lack of prestige of work, as well as a lack of interest in activities in this industry (Figure 4).

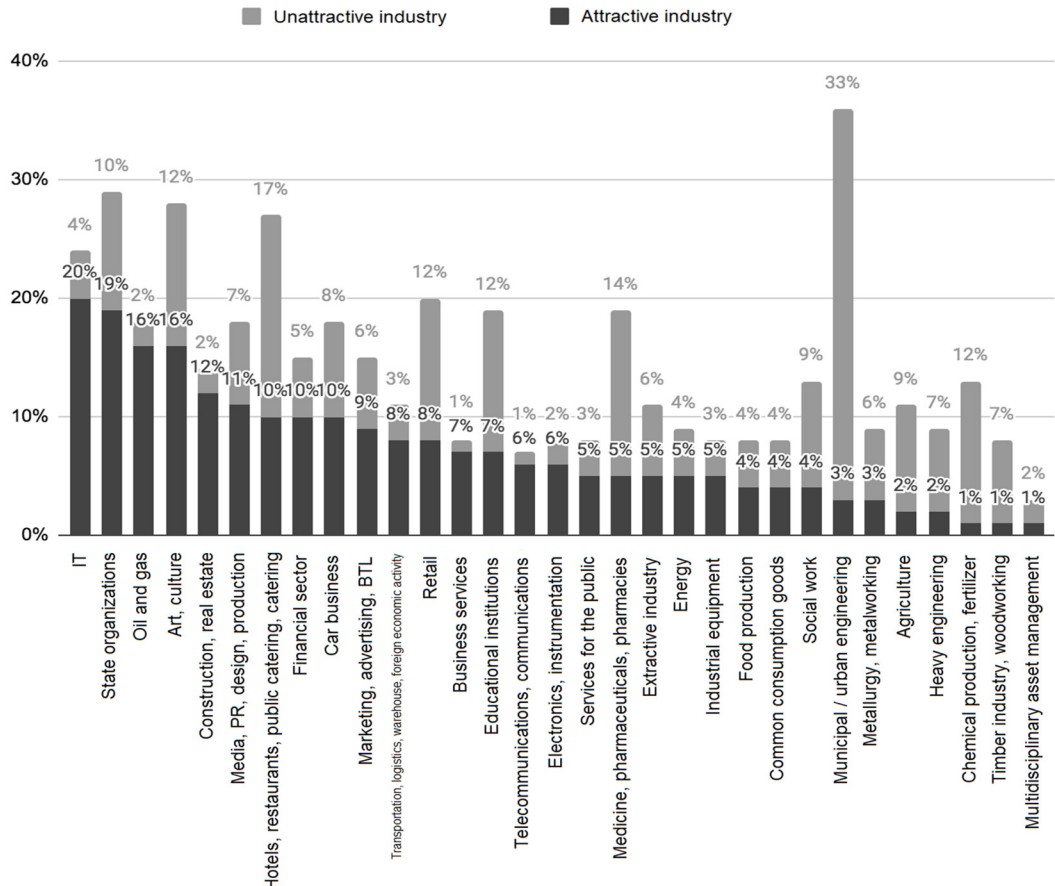

**Figure 4.** Distribution of industries according to their attractiveness among applicants (Compiled by the authors based on materials of Almanac HeadHunter (2017)).

IT industry turned out to be not only the most attractive for applicants, but also became the industry whose attractiveness is due to the largest number of different factors (when answering the question about the reasons for the attractiveness of the industry, respondents chose the largest number of different answer options). More than half of the respondents chose the prospects, dynamic development and good salaries as the reasons for the attractiveness of IT industry, however, factors such as stability, variety of activities or flexible hours turned out to be important for a significant number of applicants.

Work in government organizations attracts applicants for a completely different reason: 79% would like to work there because of the stability of this industry. The oil and gas industry is attractive mainly because of its high salaries, although factors such as prestige, stability and ability to work for a large company are also important for applicants. High salaries are also the main reasons for the attractiveness of industries such as the financial sector or car business (Almanac HeadHunter 2017).

The art and culture industry attracts applicants mainly due to the variety of activities: this option was chosen by 77% of respondents (Table 1). A variety of job responsibilities is also a major factor in the attractiveness of industries such as "Media, PR, Design, Production", "Marketing, Advertising, BTL" and "Hotels, Restaurants, Public Catering, Catering" (for the first two industries, high salaries, and perspective is important) (Almanac HeadHunter 2017).

**Table 1.** Reasons why industries are attractive to applicants (Almanac HeadHunter 2017).

| | IT | State Organizations | Oil and Gas | Art, Culture | Construction, Real Estate | Media, PR, Design, Production | Hotels, Restaurants, Public Catering, Catering | Financial Sector | Car Business | Marketing, Advertising, BTL |
|---|---|---|---|---|---|---|---|---|---|---|
| Perspective area | 64% | 25% | 41% | 21% | 41% | 45% | 31% | 40% | 30% | 49% |
| Good salaries | 57% | 30% | 87% | 10% | 55% | 41% | 29% | 69% | 42% | 48% |
| I have a corresponding education | 44% | 26% | 17% | 32% | 49% | 32% | 27% | 50% | 33% | 30% |
| Dynamically developing industry | 60% | 5% | 25% | 11% | 27% | 33% | 27% | 22% | 21% | 40% |
| Variety of activities | 40% | 14% | 14% | 77% | 34% | 60% | 46% | 22% | 26% | 56% |
| Career prospects | 36% | 39% | 40% | 13% | 36% | 35% | 26% | 45% | 26% | 38% |
| Prestige of the word in industry | 33% | 34% | 60% | 19% | 29% | 41% | 11% | 51% | 27% | 36% |
| Flexible working hours | 27% | 5% | 4% | 25% | 10% | 25% | 26% | 7% | 10% | 20% |
| Industry stability | 25% | 79% | 55% | 5% | 32% | 9% | 18% | 38% | 26% | 16% |
| Opportunity to work in a large company | 24% | 17% | 46% | 11% | 26% | 26% | 18% | 34% | 30% | 29% |
| Presence of bonuses and awards | 14% | 22% | 35% | 5% | 19% | 14% | 20% | 36% | 18% | 21% |
| Other | 2% | 1% | 0% | 7% | 1% | 3% | 5% | 1% | 4% | 2% |
| I am at a loss to answer | 1% | 2% | 1% | 4% | 1% | 2% | 5% | 2% | 5% | 3% |

Thus, the main factors for the attractiveness of work in a particular industry are high salaries, prospects of the industry, its stability and variety of activities. These factors can affect the willingness of employees to find a job in a company from the industry and the desire of employees already working in the industry to keep their jobs.

Low salaries and lack of interest are the main reasons for unattractiveness in almost all industries. Other common disincentives are the lack of relevant experience and education among applicants or work outside their specialty. In this case, they can be taken out of the scope of consideration, since they relate rather to the qualities of the applicants, and not to their assessment of a particular industry. For example, the unattractiveness of such "Medicine, Pharmaceuticals" is caused almost exclusively by the poor idea of applicants about the nature of the work, lack of knowledge, experience, education and interest in the industry. This suggests that while the industry may attract fewer applicants, it will remain quite attractive for its workers (Green et al. 2015).

The retail industry became the leader in terms of lack of interest from applicants (other distinctive details of work in this industry were high staff turnover, uncomfortable working conditions and increased conflict). Low salaries were most often embarrassing for applicants in industrie,s such as the public sector and education. One in four applicants who noted the public sector as unattractive, indicated negative reviews from friends and acquaintances as one of the reasons for choosing, which indicates a widespread negative experience in this industry. A significant number of applicants (38%) associate the chemical industry with uncomfortable or difficult, and even harmful, working conditions (Table 2).

**Table 2.** Reasons why industries are unattractive for applicants (Almanac HeadHunter 2017).

| | Municipal/Urban Engineering | Hotels, Restaurants, Public Catering, Catering | Medicine, Pharmaceuticals, Pharmacies | Art, Culture | Educational Institutions | Chemical Production, Fertilizers | Retail | State Organizations | Agriculture | Social Work |
|---|---|---|---|---|---|---|---|---|---|---|
| I do not like it, it is not interesting for me | 51% | 53% | 31% | 42% | 44% | 40% | 68% | 50% | 48% | 58% |
| Low salaries | 50% | 22% | 16% | 28% | 61% | 6% | 29% | 60% | 34% | 24% |
| Work outside the specialty | 30% | 38% | 46% | 39% | 31% | 34% | 32% | 19% | 34% | 31% |
| Conflict work | 40% | 25% | 5% | 4% | 26% | 2% | 33% | 35% | 2% | 23% |
| Not a prestigious industry | 33% | 17% | 4% | 11% | 22% | 6% | 26% | 20% | 30% | 16% |
| No knowledge, experience, appropriate education | 24% | 28% | 61% | 45% | 32% | 49% | 17% | 15% | 34% | 29% |
| Uncomfortable, difficult working conditions | 24% | 22% | 7% | 3% | 24% | 38% | 33% | 28% | 33% | 9% |
| High staff turnover (chaos) | 23% | 30% | 3% | 4% | 10% | 2% | 36% | 21% | 4% | 8% |
| Negative reviews from friends/acquaintances | 22% | 11% | 4% | 3% | 14% | 5% | 16% | 26% | 4% | 11% |
| I have little idea of work | 19% | 21% | 36% | 38% | 14% | 36% | 11% | 12% | 28% | 31% |
| No learning opportunity | 12% | 7% | 11% | 7% | 5% | 7% | 9% | 12% | 8% | 7% |
| Unstable work/crisis in the industry | 12% | 13% | 3% | 13% | 12% | 3% | 20% | 13% | 17% | 14% |
| Territorial inconvenience (far from home) | 1% | 2% | 1% | 1% | 1% | 5% | 2% | 3% | 19% | 1% |
| Other | 1% | 1% | 1% | 0% | 1% | 3% | 1% | 3% | 0% | 2% |
| I am at a loss to answer | 4% | 3% | 3% | 4% | 3% | 5% | 2% | 4% | 4% | 5% |

## 5. Conclusions

The data of opinion polls show that the choice of profession or work area has a great influence on the labor mobility of applicants. Specialists in some areas find highly susceptible to job change sentiments, even though their income is high: for example, salespeople, among whom one in five would change jobs if not for the difference in salaries. There is a significant number of applicants on the Russian labor market who want to change their profession, but do not do it for the sake of maintaining the current level of income. Most of these applicants are not only in the field of sales, but also in the areas related to finance and accounting. Although Russian applicants continue to be more guided by income in their choice of profession or industry, other labor values also play an important role: among them there are dissatisfaction with work activities, lack of self-realization or social benefit from work (Almanac HeadHunter 2017).

The share of those wishing to change their profession is one and a half times or more higher among young people than among older specialists. This can be explained both by the evolution of work values among younger applicants and by the factor of maturation: it is possible that the share of those wishing to change their profession decreases with age due to the fact that some applicants make a transition to another profession or get used to the current professional field. Most likely, the situation is influenced by all the factors described above.

The main reason for the desire to change the professional field, in addition to dissatisfaction with the financial situation, is the desire to try something new and to realize some of your hobbies. It seems logical that the professional field "Art, Culture" is attractive for applicants (if we exclude the material factor, significant number of applicants would like to work there). The second most popular area is "Staff Management" (perhaps, this is how the desire of applicants to work with people manifests itself), although 6% of its employees would like to change their profession, if not for the material factor.

The analysis of the attractiveness of work in various industries demonstrates the importance of the material factor: the most attractive industries for applicants turned out to be such because of high salaries. However, for some industries, completely different factors turn out to be decisive. Thus, a significant number of applicants would like to work in the public sector because of the stability, and in the field of art and culture because of the diversity of activities. The oil and gas industry also attracts a significant number of applicants, not only because of salaries, but also because of the prestige of the companies. However, the most attractive industry in the Russian labor market is IT industry, influenced by both high salaries in the industry and its prospects and dynamic development.

The most unattractive industry in the Russian labor market is municipal or urban engineering industry, which is influenced not only by low salaries or lack of interest from applicants, but also by high conflicts and lack of professional prestige. The industries "Hotels, Restaurants, Public Catering, Catering", "Art, Culture", and "Government Organizations" were included not only in the rating of the most attractive industries, but also in the rating of the least attractive industries (which may be one of the reasons for the high turnover of staff).

The main factors of the attractiveness of work in a particular industry for applicants are high salaries, the prospects of the industry, its stability and a variety of activities. Low salaries and lack of interest are the main internal reasons for the unattractiveness of industries (the main external reasons are lack of interest or relevant experience among applicants). All these factors also have a significant impact on the labor mobility of applicants, and their unequal representation in various industries makes some industries more prone to high staff turnover than others. Taking these factors into account can help to increase the effectiveness of staff turnover management in enterprises operating in these industries.

As a result of the research carried out by the authors using the methods of mathematical statistics, the hypotheses were confirmed that the majority of employees make their choice of their profession quite consciously, taking into account the prospects for additional development. At the same time, it turned out that not all workers are completely satisfied with their profession; there is a significant category of staff who do not quite understand the potential for demand for their specialty in the labor

market; a special category of workers is represented by young people who are most prone to frequent and radical changes in their professional life.

In the further development of the research direction, it is planned to further study the influence of the choice of profession on labor mobility:

- by countries of the world;
- by gender;
- exclusively for young people starting their work.

Thinking about the implications of the conclusions made outside of Russia, the authors believe that it is necessary to present a classification of labor markets by enlarged groups of countries. We put forward as a new hypothesis that the greatest correlation with Russian reality will be observed in those countries that were previously part of the USSR, taking into account many years of experience, mainly in government organizations with centralized planning. We believe that the majority of European countries, where the market economy dominates, small and medium-sized businesses are developing well and confidently, have a very different picture in the labor preferences of a significant part of the working population. It is a very big and global job to analyze regional and sectoral labor market economies in countries as different as Switzerland or Bangladesh. Each country in the world has its own unique peculiarity that requires painstaking methodical research, which sectors of the economy are most developed, what is the role of the state in supporting the priority areas of development of industry or agriculture, and, perhaps, tourism or hotel service. It is clear that our scientific research was conducted exclusively on the basis of Russian realities, which we have studied the most. However, changes in our work are bringing more and more new facts already in 2020, and the global pandemic COVID-19 has already brought such large-scale changes to the labor mobility of the population that this requires an immediate response from both scientific researchers and heads of government and commercial organizations. The noble mission of scientists is to help the population in any possible crisis with their scientific findings.

**Author Contributions:** Conceptualization, A.T. and S.N.; methodology, S.N.; software, V.K.; validation, A.T., S.N. and V.K.; formal analysis, U.S.; investigation, S.N.; resources, A.T.; data curation, A.T. and V.K.; writing—original draft preparation, S.N.; writing—review and editing, V.K.; visualization, U.S.; supervision, A.T.; project administration, S.N.; funding acquisition, U.S. All authors have read and agreed to the published version of the manuscript.

**Funding:** This research received no external funding.

**Conflicts of Interest:** The authors declare no conflict of interest.

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
