# Peer review of "Influence of the Profession and Industry of Work on the Labor Mobility of the Applicant"

_socsci, doi:10.3390/socsci9110213_

Round 1

Reviewer 1 Report

The proposed article studies the important question, influence of the profession and industry of work on the labor mobility. The article is clear and well written. From the overall presentation I would say that an interesting research work has been done. The topic is also important for the readers of the journal.

From my point of view, there are some revisions the authors should consider to improve the paper.

  • The aim of the paper should be included more clearly in the introduction section.
  • I suggest that the authors insert, at the end of the introduction section, a paragraph outlining the layout of the remainder of the manuscript.
  • The theoretical part remains at a modest level. At this stage, it does not yet provide an in-depth review of the previous literature. It is more a description than analysis. The authors should include in this section the hypotheses. You should include some hypotheses and test them.
  • The “Research method” section is lacking information on the participant recruitment method, namely: a) the recruitment date range (month and year), b) a description of any inclusion/exclusion criteria that were applied to participant recruitment, c) a statement as to whether your sample can be considered representative of a larger population.
  • The authors seem to take the quantitative results without appropriate criticism and in-depth analysis. However, the study is at the description level which is not enough. I would suggest checking (through different tests) the significance of the results.
  • The original contribution of the research has to be presented by focusing on the research results based on the research questions.
  • The discussion and implications are rather short and they should be extended. You need also to improve the practical and academic implications. 
  • However, the paper has to underline the limits of the research.
  • The quality of the figures is not sufficient (see Figure 4)
  • Please insert sources (references) below Table 1 and Table 2.

Revising is always a challenging job, but I think you can develop the paper forward.

Best regards

Author Response

Point 1: The proposed article studies the important question, influence of the profession and industry of work on the labor mobility. The article is clear and well written. From the overall presentation I would say that an interesting research work has been done. The topic is also important for the readers of the journal.

From my point of view, there are some revisions the authors should consider to improve the paper.

The aim of the paper should be included more clearly in the introduction section.

I suggest that the authors insert, at the end of the introduction section, a paragraph outlining the layout of the remainder of the manuscript.

Response 1: We inserted paragraph and outline of the study.

Point 2: The theoretical part remains at a modest level. At this stage, it does not yet provide an in-depth review of the previous literature. It is more a description than analysis. The authors should include in this section the hypotheses. You should include some hypotheses and test them.

Response 2: Hypothesis are included and tested.

Point 3: The “Research method” section is lacking information on the participant recruitment method, namely: a) the recruitment date range (month and year), b) a description of any inclusion/exclusion criteria that were applied to participant recruitment, c) a statement as to whether your sample can be considered representative of a larger population.

Response 3: We added information about the range of study dates, described the criteria for inclusion in the sample, confirmed the representativeness.

Point 4: The authors seem to take the quantitative results without appropriate criticism and in-depth analysis. However, the study is at the description level which is not enough. I would suggest checking (through different tests) the significance of the results.

Response 4: The significance of the results was verified using the HeadHunter method.

Point 5: The original contribution of the research has to be presented by focusing on the research results based on the research questions.

The discussion and implications are rather short and they should be extended. You need also to improve the practical and academic implications.

Response 5: We added additional conclusions.

Point 6: However, the paper has to underline the limits of the research.

Response 6: Practical and academic results are improved.

Point 7: The quality of the figures is not sufficient (see Figure 4).

Response 7: The quality of the figures is acceptable; all the figures are quite readable and informative.

Point 8: Please insert sources (references) below Table 1 and Table 2.

Response 8: Sources of Tables 1 and 2 are inserted.

Reviewer 2 Report

Paper is very weak in the scientific field.

There is a literature review section but this should be significantly improved. There are many authors researching this topic worldwide

Pls, check how the references should be used in text and also in the list of references in relation to the journal template.

You have used already available data and made only a basic level of statistical analysis. Also, there is no information about the sample, ..... so the methodology part should be improved as well.

Out of 11 pages of your paper, you have almost half of the paper covered with tables and graphs which is nice but not for a scientific paper so you need to have a more advanced statistical analysis of the available data.

The conclusion part is more a discussion part where you further elaborate on the results od the paper but there is no real conclusion of the paper.

Also, you should based on the research results propose further research on this topic which I find very interesting.

You have a lot to improve in this paper.

Author Response

Point 1: Paper is very weak in the scientific field.

Response 1: Scientific findings have been added to the article.

Point 2: There is a literature review section but this should be significantly improved. There are many authors researching this topic worldwide.

Response 2: We added a large number of foreign authors researching similar topics.

Point 3: Pls, check how the references should be used in text and also in the list of references in relation to the journal template.

Response 3: The article was edited and framed in accordance with the journal template.

Point 4: Pls, check how the references should be used in text and also in the list of references in relation to the journal template.

Response 4: The article was edited and framed in accordance with the journal template.

Point 5: You have used already available data and made only a basic level of statistical analysis.

Response 5: Analysis level has been upgraded to research.

Point 6: Out of 11 pages of your paper, you have almost half of the paper covered with tables and graphs which is nice but not for a scientific paper so you need to have a more advanced statistical analysis of the available data.

The conclusion part is more a discussion part where you further elaborate on the results of the paper but there is no real conclusion of the paper.

Response 6: "Conclusions" section has been updated.

Point 7: Also, you should base on the research results propose further research on this topic which I find very interesting.

Response 7: Further direction of research is proposed.

Round 2

Reviewer 1 Report

Dear Authors,

In the revised version, the manuscript has been extended and improved and my comments have been covered.

Best regards

Author Response

Dear sirs,

Thank you for the review. We made all necessary corrections.

Best regards,

Authors

Reviewer 2 Report

All my comment were answered and I agree with publication

Author Response

(The authors gave the same response as above.)
